# A Serial Mediation Model of Insecure Attachment and Psychological Distress: The Role of Dispositional Shame and Shame-Coping Styles

**DOI:** 10.3390/ijerph20043193

**Published:** 2023-02-11

**Authors:** Chiara Remondi, Giulia Casu, Camilla Pozzi, Francesco Greco, Paola Gremigni, Agostino Brugnera

**Affiliations:** 1Department of Psychology, Sapienza University of Rome, Via dei Marsi 78, 00185 Rome, Italy; 2Department of Psychology, University of Bologna, Viale Berti Pichat 5, 40127 Bologna, Italy; 3Independent Researcher, Via Stefano Gobatti 5, 40137 Bologna, Italy; 4Department of Human and Social Sciences, University of Bergamo, P.le S. Agostino 2, 29129 Bergamo, Italy

**Keywords:** attachment, dispositional shame, shame-coping, psychological distress, mediation

## Abstract

Shame is an intense, difficult to regulate, self-conscious emotion that predicts aspects of poor psychological functioning and is also strongly related to early relationships. Attachment insecurities, which constitute non-specific risk factors for psychological maladjustment, have been associated with an individual’s tendency to experience shame. In this study, we sought to examine the serial mediating roles of dispositional shame and shame-coping styles (i.e., attack other, attack self, withdrawal and avoidance) in the association between anxious and avoidant attachment, and psychological distress. Using a cross-sectional design, self-reported data were collected. The study sample included 978 respondents (57% female) with a mean age of 32.17 ± 13.48 years. The results of the path analysis indicated that both attachment dimensions were sequentially associated with dispositional shame and then with the attack self shame-coping style, which was, in turn, positively related to psychological distress. Further, attachment insecurities were sequentially associated with dispositional shame and then with the avoidance shame-coping style, which was, in turn, negatively related to psychological distress. The model was gender invariant, suggesting that the serial mediation worked in a similar way for men and women. The practical implications of these findings are discussed.

## 1. Introduction

Attachment theory has been proposed as a useful framework for understanding both the development of a healthy and effective self [1] and the etiology of many mental disorders [2]. According to attachment theorists, individuals who experience repeated early interactions with inconsistent and/or unresponsive caregivers will develop an insecure, anxious or avoidant attachment, characterized by negative internal working models (IWMs) of self and others [3]. Anxiously attached individuals experience recurrent difficulties in trusting others, a strong desire to become closer to others, fear of abandonment or exclusion and generalized difficulty to down-regulate their negative emotions [3,4,5,6]. In contrast, avoidantly attached individuals experience fear of intimacy, discomfort with closeness, a strong need for independence, control and autonomy in intimate relationships, and tend to deactivate and suppress their emotions during everyday life and in romantic relationships [3,4,5,6].

Previous studies showed that attachment insecurity is an overall risk factor that non-specifically contributes to the development of a variety of mental disorders and symptoms [3,7], including psychological distress, depression and anxiety [8,9]. Indeed, as they favor maladaptive coping strategies and negative self-perceptions, insecure IWMs interfere with personality formation, affect emotional and cognitive development and foster interpersonal difficulties that persist throughout adulthood [3,7]. Over the past decades, the interest in examining the potential mediators of the relationship between insecure attachment and psychological distress has progressively increased (e.g., [10,11,12]). Once identified, these mediators, together with the IWMs [13], might be targeted by mental health practitioners to help individuals relieve their symptoms. Given their purported role in influencing the onset and maintenance of psychological distress [14,15,16,17] and their strong relationships with attachment representations [18,19,20], two promising mediators are shame and shame-coping styles.

Shame is an intense, self-conscious emotion elicited when the individual is afraid of being judged, ridiculed or denigrated, which involves feelings of worthlessness, powerlessness and inferiority, along with the desire to conceal their deficiencies [21,22]. Although shame might be an emotion that all people feel under some circumstances (i.e., situational shame), repeated emotional experiences of shame may bring people to reach an understanding of themselves as shamefully inadequate in many areas of life (i.e., dispositional shame) [23].

Shame has been predominantly linked to negative consequences, such as depression and social anxiety [21,24]. However, evolutionary perspectives have focused on the positive aspects of shame (for a review, see [25]). In fact, shame develops in order to guarantee the continuity of the individual in his or her social group through the activation of positive interpersonal behaviors aimed at preserving the relationship with others and proving oneself as acceptable to others [26]. Thus, shame-avoidance is a priority for every social group member, as it both increases the chance of survival and provides better access to valuable resources (e.g., food) [27].

Shame is currently considered a multidimensional construct. Indeed, individuals could experience shame for behavioral (e.g., doing or saying something wrong, failing in competitive situations), characterological (e.g., personal habits, relational styles, personality characteristics, abilities) and bodily aspects of self (e.g., feeling ashamed of one’s body or parts of it) [21]. When shame is not linked to social interactions or specific events, it loses its adaptive value and becomes a dispositional trait characterized by an unacceptable, inadequate and defective image of the self [24].

There is an increasing interest in psychological research on dispositional shame and its association with psychological functions [28]. For example, an individual’s tendency to frequently experience shame has been associated with lower self-esteem [28,29] and higher hostility and psychological distress [14,28]. Furthermore, consistent findings suggest that dispositional shame is an important predictor of the onset and maintenance of psychiatric disorders, such as depression [27,30] and personality disorders [31,32].

Shame is also one of the most challenging emotions to regulate [29]. Once experienced, individuals cope with it in different ways. For example, some tend to magnify the impact of shame, while others tend to minimize it. The way in which people cope with shame has important implications. It has been suggested that psychological distress could be explained by how one copes with shame, rather than being an effect of shame per se [28]. Based on clinical observations, Nathanson [33] proposed the so-called Compass of Shame model. According to it, individuals usually adopt four different maladaptive coping styles, considered as relatively stable dispositions, in order to manage shame-related attributions and emotions, namely attack self, attack other, withdrawal and avoidance. A commonly adopted shame-coping style is attack self. In front of shameful situations, the individual acknowledges shame, accepts it as a valid emotion and expresses rageful and hateful feelings toward her or himself [15]. In contrast, individuals who adopt the attack other coping style neither validate shame nor recognize it; they blame and attack others to diminish their shameful experience. In the withdrawal coping style, shame is acknowledged and accepted. However, the person attempts to escape or hide from the shameful situation [15]. Finally, individuals who adopt an avoidant style tend not to acknowledge shame, try to distance themselves from it and transform the experience of shame into a neutral or positive one, for example, by joking about it [15,16,33].

The shame-coping styles described above differ in the extent to which shame is subjectively experienced and they are usually maladaptive due to their impact on people’s mental health and well-being [34]. Few studies have investigated the contribution of specific shame-coping styles on psychological adjustment. For instance, Elison and colleagues [15] found that attack self and withdrawal were moderately-to-strongly linked to lower self-esteem, higher psychopathology and depressive symptoms, whereas attack other was moderately and positively associated with interpersonal anger and psychopathy [35]. In summary, maladaptive shame regulation strategies can result in both internalizing (e.g., psychological distress) and externalizing (e.g., aggression or hostility) the symptoms [15,16,28,29].

Shame is strongly related to early relationships. It occurs early in life in response to perceived rejection or separation from attachment figures [36]. Attachment theory provides an ecologically and scientifically-sound framework to understand how early experiences of shame may lead to psychopathology [3]. Indeed, early interactions with attachment figures that are characterized by recurrent experiences of shame could be organized into negative mental schemes (i.e., IWMs) of the self as worthless, undesirable or defective, and of others as threatening, powerful, superior, hostile and judgmental [3,19,20]. More in detail, because anxiously attached individuals hold negative views of the self, they might be more prone to experience feelings of shame when they do not appear “perfect” and when they are unable to create a sense of belonging with others [3,37]. Conversely, avoidantly attached individuals, who have a negative view of others and a positive view of themselves, may be more likely to withdraw from others to avoid feelings of shame [10,18]. Insecure attachment, especially anxious attachment, is also a strong predictor of women’s experiences of body shame and surveillance [38]. To the best of our knowledge, no previous study has examined how insecure attachment dimensions may be differently related to the shame-coping styles. However, one may argue that while anxiously attached individuals tend to own a negative self-image and may attack themselves as a way of not being rejected by others [3,39], individuals high on attachment avoidance may prefer to use strategies that suppress the attachment system in order to maintain independence from others, such as an avoidant shame-coping style [40].

If insecure attachment non-specifically contributes to the onset and development of psychopathology [7], then other psychological variables should mediate this relationship [10,11,12]. Identifying such mediators could have important implications for clinical practice. However, to date, no studies have investigated whether insecure attachment may lead to higher levels of psychological distress through shame. Therefore, in this cross-sectional study, we tested a serial mediation model in which dispositional shame and the shame-coping styles mediated the associations of insecure attachment with psychological distress (see Figure 1 for a graphical depiction of the hypothesized model). Based on the aforementioned considerations, we hypothesized the following.

**Hypothesis 1 (H1).** 
*Higher attachment insecurity will be positively associated with dispositional shame, shame-coping styles and psychological distress. In particular, attachment anxiety will be positively related to the attack self shame-coping style, whereas attachment avoidance will be positively related to withdrawal and the avoidance shame-coping styles. Dispositional shame will be positively associated with all four shame-coping styles and psychological distress. The shame-coping styles will be positively associated with psychological distress.*


**Hypothesis 2 (H2).** 
*Dispositional shame will mediate the direct association of attachment insecurity with psychological distress. Specifically, we hypothesize that higher levels of attachment anxiety and attachment avoidance will be associated with higher levels of dispositional shame, which, in turn, will be associated with higher psychological distress.*


**Hypothesis 3 (H3).** 
*The shame-coping styles will mediate the direct association of attachment insecurity with psychological distress. Specifically, we expect that higher attachment anxiety will be related to higher levels of the attack self shame-coping styles, which, in turn, will be linked to higher psychological distress. For attachment avoidance, we hypothesize that it will be associated with higher levels of withdrawal and the avoidance shame-coping styles, which, in turn, will be related to higher distress.*


**Hypothesis 4 (H4).** 
*Dispositional shame and shame-coping styles will be serial mediators in the relation between attachment insecurity and psychological distress. We posit that higher attachment anxiety will be associated with higher distress via higher dispositional shame and then greater use of the attack self shame-coping styles and a higher attachment avoidance will be linked with higher distress via higher dispositional shame and the greater use of withdrawal and the avoidance shame-coping styles.*


Numerous studies examined the gender differences in shame. While some studies found that females were more prone to experience and express feelings of shame [41], others studies have shown that shame is equally experienced across gender [42,43]. For instance, a recent study [42] showed that gender did not affect the pathways linking shame and the shame-coping styles to internalizing/externalizing the symptoms. To further evidence the role of gender in the link between shame and psychological distress, we tested whether the proposed serial mediation model was invariant across gender.

Support for the proposed serial mediation model may provide preliminary evidence of the role of shame on psychological distress within the framework of attachment theory, as well as valuable insights for psychotherapeutic and counseling practice.

## 2. Materials and Methods

### 2.1. Participants and Procedure

This study adopted a cross-sectional design. Data were collected using a web-based survey. The inclusion criteria to complete the survey were an age above 18 years and being a native Italian speaker. The eligible participants were sent an e-mail by the students attending graduate courses at the University of Bologna, containing a brief description of the study and a link to the web-based survey. Each student sent an e-mail to 20 persons in his or her mailing list and a total of 60 students contributed to the data collection. The participants were recruited on a voluntary basis. Those who agreed to participate were redirected to a webpage that detailed the aims of the research and the names and contact information of the researchers. A separate consent screen appeared before the respondent could access the survey requiring participants to click on an “I agree” button before moving forward into the survey. This study was approved by the Bioethics Committee of the University of Bologna (Prot. 43743, 15 March 2018) and was conducted in accordance with the ethical standards for the treatment of human experimental volunteers.

### 2.2. Measures

The Italian version of the Attachment Style Questionnaire (ASQ) [44,45] is a self-reported measure of the attachment dimensions in general (i.e., non-romantic) relationships, which includes 40 items scored on a six-point scale (1 = totally disagree; 6 = totally agree). The ASQ contains five subscales: confidence in the self and others (eight items, e.g., “I feel confident that other people will be there for me when I need them”), need for approval (seven items, e.g., “It’s important to me to avoid doing things that others won’t like”), preoccupation with relationships (eight items, e.g., “I worry a lot about my relationships”), discomfort with closeness (10 items, e.g., “I find it difficult to depend on others”) and relationships as secondary (seven items, e.g., “Achieving things is more important than building relationships”). It is possible to compute two second-order dimensions, namely attachment anxiety and attachment avoidance, summing the need for approval and preoccupation with the relationships subscale scores for attachment anxiety, and the preoccupation with relationships and the discomfort with closeness subscale scores for attachment avoidance. Higher scores in these dimensions indicated higher levels of attachment anxiety and avoidance, respectively. The ASQ evidenced a good construct validity and internal reliability [44,45]. In the present study, Cronbach’s *α*s were 0.83 for anxiety and 0.80 for avoidance.

The Italian version of the Experience of Shame Scale (ESS) [21,46] is a 25-item questionnaire designed to assess dispositional shame. The ESS is composed by three subscales, namely characterological shame (12 items, e.g., “Have you felt ashamed of any of your personal habits?”), behavioral shame (nine items, e.g., “Do you feel ashamed when you do something wrong?”) and bodily shame (four items, e.g., “Have you felt ashamed of your body or any part of it?”). The sum of the three components provides an overall index, with higher scores indicating higher levels of dispositional shame. The items are scored on a four-point scale (1 = not at all; 4 = very much). The ESS evidenced good psychometric properties [21]. In the present study, Cronbach’s *α* for the overall score was 0.94.

The Compass of Shame Scale (CoSS) [15] is a 48-item self-reported inventory of coping styles triggered in reaction to a shaming event. The CoSS includes four 12-item subscales, namely attack other (e.g., “When I feel others think poorly of me, I want to point out their faults”), attack self (e.g., “In competitive situations where I compare myself with others, I criticize myself”), withdrawal (e.g., “At times when I am unhappy with how I look, I keep away from other people”) and avoidance (e.g., “When an activity makes me feel like my strength or skill is inferior, I act as if it isn’t so”). The items are rated on a five-point scale (0 = never; 4 = almost always). Higher sum scores in each subscale indicate higher use of that specific shame-coping style. Previous research evidenced good psychometric properties for the CoSS [15]. In the present study, we used an Italian version of the CoSS obtained through a back-translation process using two independent bilingual translators [47]. Cronbach’s *α*s were 0.85 for attack other, 0.92 for attack self, 0.90 for withdrawal and 0.80 for avoidance.

The Italian version of the Kessler Psychological Distress Scale (K10) [48,49] is a 10-item self-reported measure of anxiety and depressive symptoms experienced during the last four weeks (e.g., “During the last 30 days, about how often did you feel depressed?”). The items are rated on a five-point scale (1 = none of the time; 5 = all of the time). An overall score was computed by summing up all the items, with higher scores indicating higher levels of psychological distress. The K10 showed good psychometric properties [48,49]. In the present study, Cronbach’s *α* was 0.88.

### 2.3. Data Analysis

An examination of the univariate and multivariate outliers was conducted using *z*-scores with a cut-off of 3.29 and the Mahalanobis distance with a *p*-value of 0.001, respectively [50]. Sixteen cases were identified as multivariate outliers and were thus excluded, leading to a final sample of 978 participants. Subsequent preliminary analyses included the assessment of univariate normality through the skewness and kurtosis values < |1| [50]. To test for the multivariate normality, Henze–Zirkler’s, Mardia’s and Royston’s multivariate normality tests were used. The tolerance and variance inflation factor (VIF) were computed to exclude the multicollinearity between the predictors in the model, with the values of the tolerance of >0.10 and the values of the VIF of <5 considered as acceptable [51]. Only the sociodemographic variables that were correlated with the mediators or outcome at *r* ≥ |0.30| were included in the model as the covariates [52].

The hypothesized mediation model was tested within the structural equation modeling (SEM) framework and all the parameters were estimated using the maximum likelihood with standard errors robust to non-normality (MLR) estimator [53]. In particular, we tested the serial mediation model, which incorporated the two serial mediators’ dispositional shame and shame-coping styles to assess the direct and indirect (i.e., specific and serial) associations of attachment insecurity with psychological distress. The model fit was evaluated using the following criteria: a root mean square error of approximation (RMSEA) of ≤0.06; a comparative fit index (CFI) and a Tucker–Lewis index (TLI) of ≥0.95 [54], and a standardized root mean squared residual (SRMR) of ≤0.05 [54]. In the case of a non-optimal fit, the modification indices were examined to find the most parsimonious changes to the model to achieve an acceptable fit. We computed the indirect associations using the bias-corrected bootstrapping method with 5000 replications and a 95% confidence interval (CI). An indirect effect was considered significant when the 95% CI did not include zero [55]. To test if the model was invariant across gender, we performed a multi-group analysis with gender as the grouping variable. We first estimated a two-group fully unconstrained model in which all pathways were allowed to vary across gender, then estimated a constrained model in which these parameters were set to be equal across gender [46]. The invariance was established if the CFI difference (ΔCFI) between the nested models was higher than the recommended threshold of 0.01 [56].

Pearson’s *r* (for preliminary analyses) and the path coefficients (for the SEM) were considered as follows: small correlations by 0.10 < *r* < 0.30, medium correlations by 0.30 < *r* < 0.50 and large correlations by *r* > 0.50 [57]. All the statistical tests were two-tailed and a *p* value < 0.05 was considered statistically significant. The SEM analyses were conducted using Mplus 8.4 [53]. All the other analyses were conducted using IBMS SPSS 28.

## 3. Results

### 3.1. Sample Characteristics

The participants in the final sample (*n* = 978) were 57% female (*n* = 557) with a mean age of 32.17 ± 13.48 years (ranging from 18–84). As for their educational level, 475 had a high school diploma or less (48.6%) and 503 held a bachelor’s degree or higher (51.4%). Forty-four percent were active community workers (*n* = 431), 45% were university students (*n* = 440) and 10.9% were unemployed, retired or housewives (*n* = 107).

### 3.2. Preliminary Analyses

The assessment of the univariate normality indicated that all the study variables had a univariate normal distribution, with both skewness and kurtosis < |1| (See Appendix A). The study variables had a multivariate normal distribution according to the Henze–Zirkler’s statistic (HL = 1.58, *p* = 0.32). However, Mardia’s and Royston’s statistics indicated significant departures from the multivariate normality (all *p*s < 0.001). The correlations among the study variables were all significant and positive, with small to large effects. The tolerance and VIF were in the 0.33–0.82 and 1.22–2.99 range, respectively, excluding the multicollinearity problems between independent variables of our model. Age correlated > |0.30| with dispositional shame, hence, it was included as a covariate in the mediation model. The descriptive statistics and Pearson’s correlations are reported in Table 1.

### 3.3. Serial Mediation Model

The results of the SEM analysis revealed that the hypothesized serial mediation model did not show a satisfactory fit, *χ^2^*(7) = 74.409, *p* < 0.001, CFI = 0.980, TLI = 0.908, RMSEA = 0.099 (90% CI 0.080–0.120), SRMR = 0.050. An inspection of modification indices suggested to add the effect of age on attachment anxiety. The model was reanalyzed and yielded a good fit to the data, *χ^2^*(6) = 28.086, *p* < 0.001, CFI = 0.994, TLI = 0.964, RMSEA = 0.061 (90% CI = 0.040–0.085), SRMR = 0.017. The standardized path coefficients are presented in Figure 2. The total, specific indirect and serial indirect effects along with their 95% bias-corrected CIs are displayed in Table 2.

Both attachment insecurities were directly and positively associated with psychological distress with small effects.

With regard to specific indirect effects, both the attachment insecurity dimensions had significant specific indirect effects on psychological distress through dispositional shame. Higher attachment anxiety and attachment avoidance were associated with higher dispositional shame, with medium and small effects, respectively, and thus, with moderately higher psychological distress. The specific indirect effects on psychological distress through the shame-coping styles differed between the attachment insecurity dimensions. Attachment anxiety had a significant specific indirect effect on psychological distress through the attack self shame-coping style and attachment avoidance through the avoidance shame-coping style. Thus, higher attachment anxiety was associated with moderately higher levels of attack self shame-coping, and thus, with slightly higher psychological distress, whereas higher attachment avoidance was linked to higher avoidance shame-coping, and thus, with lower psychological distress with small effects. Both attachment insecurities were linked with small-to-medium effects to higher levels of the attack other and withdrawal shame-coping styles, which nonetheless were unrelated to psychological distress.

With regard to serial indirect effects, higher attachment anxiety and avoidance were sequentially associated first with higher dispositional shame and then with a higher attack self shame-coping style, which was, in turn, related to slightly higher psychological distress. Higher attachment anxiety and avoidance were also sequentially associated, first with higher dispositional shame and then with a higher avoidance shame-coping style, which was, in turn, related to slightly lower psychological distress.

Overall, the variance accounted by the posited model was 44% for dispositional shame, 33% for attack other, 48% for attack self, 49% for withdrawal, 16% for avoidance shame-coping style and 47% for psychological distress.

### 3.4. Multi-Group Analysis

The results of the multi-group analysis showed that the fit of the model in which all the path coefficients were constrained to be equal across gender was adequate and not worse than the fit of the unconstrained model (ΔCFI = 0.006), supporting gender invariance. See Appendix A for Pearson’s correlations among the study variables by gender and the fit indices of the nested models (Appendix A).

## 4. Discussion

In this study, we sought to examine the serial mediating roles of dispositional shame and the shame-coping styles in the relation between insecure (i.e., anxious and avoidant) attachment dimensions and psychological distress. First, we found that both the attachment insecurity dimensions had significant specific indirect effects on psychological distress through dispositional shame. We also found that the specific indirect effects on psychological distress through the shame-coping styles differed between the attachment insecurity dimensions. Attachment anxiety had a significant specific indirect effect on psychological distress through the attack self shame-coping style and attachment avoidance through the avoidance shame-coping style. Second, we found four serial indirect effects of attachment insecurities on psychological distress through dispositional shame first and then the shame-coping styles. Attachment anxiety and avoidance were sequentially associated, first with higher dispositional shame and then with higher attack self shame-coping, which was, in turn, positively related to psychological distress. Attachment anxiety and avoidance were also sequentially associated, first with higher dispositional shame and then with higher avoidance shame-coping style, which was, in turn, negatively related to psychological distress.

In accordance with H1, attachment anxiety and attachment avoidance had a positive association with dispositional shame, with moderate and small effects, respectively. This result is in line with previous studies suggesting that negative self-evaluations are created from experiences of interpersonal rejection with caregivers and continue to operate internally as an IWM [58]. As a result, insecurely attached people have lower levels of self-worth and/or confidence, and this emptiness may be associated with a greater tendency to experience shame in a trait-like manner [18,59,60,61]. Moreover, individuals with an avoidant attachment tend to base their sense of self-worth on their personal competences and less on receiving social approval. Thus, they are less prone to the feelings of shame compared to anxiously attached individuals, which may explain the different magnitude of effects in our data [3]. Another explanation may be that because individuals with high attachment avoidance actively defend against or attempt to hide their vulnerable feelings, they may also under-report the acknowledgement of their feelings, such as shame [62]. We found that attachment anxiety had positive associations with the attack other, attack self and withdrawal shame-coping styles, while attachment avoidance had positive associations with the attack other, withdrawal and avoidance shame-coping styles. In other words, our findings suggest that individuals with an anxious or an avoidant attachment differ only in the adoption of one shame-coping style. While anxiously attached individuals attack themselves as a way of coping with shame, individuals with an avoidant attachment seem to prefer to be emotionally detached from or to disavow the experience of shame. More research is required to determine the distinct role of the attachment insecurity dimensions in predicting the shame-coping styles. As hypothesized, both attachment anxiety and attachment avoidance were directly associated with higher psychological distress. Consistent with previous findings, for anxiously attached individuals, the amplification and exaggeration of emotions, preoccupation with relationships and hypersensitivity to other people’s evaluations may lead them to experience greater psychological distress [63,64]. Similarly, people with an avoidant attachment are more likely to experience psychological distress because of their IWMs that emphasize the threatening and untrustworthy nature of significant others [3].

Moreover, coherent with previous studies suggesting that an individual’s tendency to frequently experience shame is accompanied by the enactment of more shame-coping styles and by greater overall psychological distress [17,28,65,66], we found that dispositional shame was positively associated with all the shame-coping styles and with psychological distress. In other words, shame as an awkward, self-disapproving and painful experience may enhance an individual’s use of maladaptive shame-coping styles, and increase their vulnerability to psychological distress.

Different from our expectations, we found that only attack self and avoidance were significantly associated with psychological distress in our mediation model. Consistent with previous research on the role of the shame-coping styles in predicting mental health difficulties, we found that attack self was associated with higher psychological distress [15,16,24,67,68]. This supports the view that attack self is a particularly maladaptive coping style for managing shame and may be a significant risk factor for subsequent psychological distress. In contrast, we found that avoidance was associated with lower psychological distress. Although surprising, this result suggests that, in some cases, an avoidance strategy may be successful in minimizing or completely preventing the feeling and awareness of shame, subsequently preventing the development of mental health difficulties, including psychological distress [29]. Another possible explanation is that avoidance helps people maintain a positive self-image and to experience less psychological distress. However, it must be acknowledged that avoidance is the hardest shame-coping style to measure via self-reporting, due to its unconscious/denial aspect. Individuals who rely more on shame-avoidance try to distance themselves from the emotional experience, and this has been linked to a lowered awareness of psychological distress [15,35]. Altogether, our findings not only validate shame as a multidimensional construct, but additionally proffers important questions on the differential impacts of the shame-coping styles on mental health outcomes.

In line with H2, dispositional shame was a significant mediator in the relationship between both the attachment insecurity dimensions and psychological distress. In other words, attachment anxiety and attachment avoidance were positively related to dispositional shame, which, in turn, was associated with higher psychological distress, as hypothesized based on the results of the preceding studies [14,15,16,28,66]. Considering the early development and relative stability of the attachment dimensions and dispositional shame, there is good reason to suspect that insecure attachment and the frequent experience of shame represent vulnerabilities to psychological distress. Dispositional shame seems to emerge from global self-evaluation in which the self is blamed and criticized. Thus, the object of negative attribution in shame is the essence of the self [27]. This entails feelings of inferiority and inadequacy that purport the negative impacts of being insecurely attached and spill into greater psychological distress [69].

In partial accordance with H3, we found that the specific indirect effects on psychological distress through the shame-coping styles differed between the attachment insecurity dimensions. Specifically, while attachment anxiety was positively related to the attack self shame-coping style, which, in turn, was linked to higher psychological distress, attachment avoidance was positively related to the avoidance shame-coping style, which, in turn, was linked to lower psychological distress. It might be argued that, although negative childhood experiences may foster shame and the development of later psychological distress, this does not necessarily mean that all individuals growing up in hostile and/or unresponsive environments are prone to experiencing shame in the same manner. According to the social mentality theory [58], individuals high in attachment anxiety have a reduced capacity for self-compassion and a reduced ability to self-soothe with a positive stance toward the self. By attacking themselves as a way of dealing with shame, they also experience heightened distress. In contrast, individuals high in attachment avoidance are likely to have experienced rejection or punishment when they have expressed distress to caregivers. Therefore, they learned to inhibit the experience of shame and the expression and recognition of emotional distress [3,7]. The small magnitude of the association between avoidance shame-coping and psychological distress, combined with its negative sign, reinforces the idea of avoidance having an adaptive quality, at least in the short-term for non-clinical populations [15,16,17,35,42,70].

In partial accordance with H4, as for the serial indirect effects, we found that attachment anxiety and avoidance were linked to higher dispositional shame, which, in turn, was associated with higher attack self and, thus, with greater psychological distress. In other words, insecure attachment seems to create a vulnerability to shame and increases the adoption of a negative shame-coping strategy, such as attacking the self, which, in turn, may lead to greater psychological distress [58,65]. We also found that attachment anxiety and avoidance were linked to higher dispositional shame, which, in turn, was associated with higher avoidance shame-coping style, and thus, with lower psychological distress. This suggests that people who exhibit an anxious or an avoidant attachment also experience greater shame and may actively distract themselves (via avoidance) from this painful feeling, thus being less likely to display distress. As Nathanson [33] argued, when individuals engage in avoidance shame-coping, they usually do it in the attempt to externalize the experience of shame, ‘transforming’ the situation into something neutral or positive. However, further studies with a more elaborative assessment of the shame-coping styles are needed to clarify the relationship between the avoidance shame-coping style and psychological distress.

The associations described so far were invariant across gender, meaning that the patterns of the relationships between attachment insecurity, dispositional shame, shame-coping styles and psychological distress were consistent across men and women. It seems that, independently of gender, both men and women with an insecure attachment may be more predisposed to experience greater dispositional shame and to enact specific shame-coping styles, which are consequently related to psychological adjustment [42].

Limitations, Future Directions and Practical Implications

This study presents some limitations that should be considered. First, this study had a cross-sectional design, therefore causal relations between the variables cannot be established. Prospective studies are needed to investigate how attachment insecurity, dispositional shame and the shame-coping styles may predict psychological distress over time. Second, we relied on self-reported measures, which, though psychometrically sound, are still vulnerable to subjective answering. Future research should corroborate the posited model using multiple raters to minimize bias due to self-reporting. A third limitation is concerned with the generalizability of the present findings. Our non-random, convenience sampling approach might have introduced a selection bias and contributed to the high proportion of students in this study. Thus, caution should be used to generalize our findings, and replication studies using random samples representative of the general population are needed. Additionally, to verify the generalizability from a community to a clinical sample, further works should explore whether the pattern of associations between our study variables is consistent across community and clinical groups.

Despite these limitations, the present study has practical implications in highlighting the relevance of improving individuals’ use of maladaptive shame-coping styles as the key factors aimed at preventing people’s psychological distress. The study findings suggest the importance of identifying the factors and resources that may help individuals with an impaired sense of self (i.e., greater dispositional shame). For example, previous research has suggested that social support and feelings of belonging are essential factors for overcoming feelings of shame [71,72]. Thus, mental health practitioners could adopt specific interventions to address dysfunctional attachment experiences and maladaptive shame-coping styles, such as psychodynamic, mentalization-based or emotion-focused psychotherapies. Bearing in mind the unhelpful role of the attack self shame-coping style observed in the current study and recent findings attesting that shame responding can be a barrier to the therapeutic alliance [73], the integration of shame interventions at early therapeutic stages aimed at generating feelings of self-reassurance, warmth and self-soothing would be beneficial.

## 5. Conclusions

On the whole, the present study was a first attempt to fill a theoretical gap by integrating broader dispositional shame and specific shame-coping styles in the relation between the attachment insecurity dimensions and psychological distress. Unfolding the role of shame on psychological distress within the framework of attachment theory opens avenues for future studies in this area and useful insights for psychotherapeutic and counseling practice.

## Figures and Tables

**Figure 1 ijerph-20-03193-f001:**
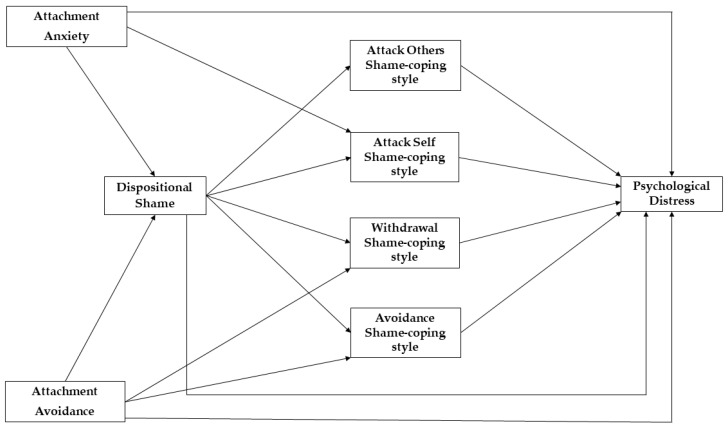
Hypothesized serial mediation model. All hypothesized associations are positive.

**Figure 2 ijerph-20-03193-f002:**
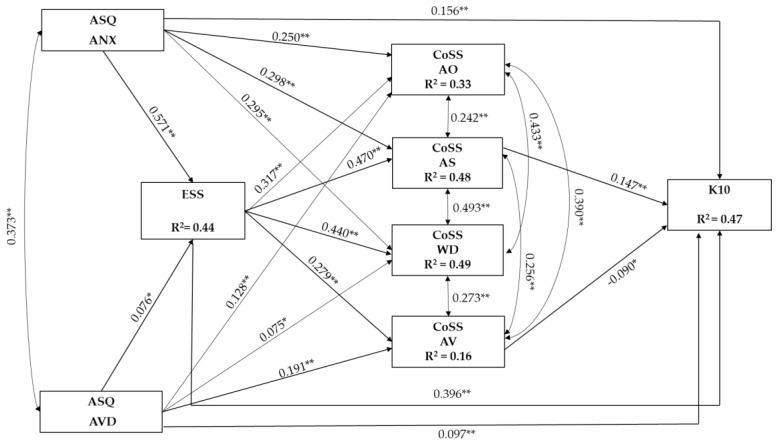
Serial mediation model (*n* = 978). Standardized parameter estimates are reported. To ease the interpretation, covariates and non-significant paths/correlations are not shown. ASQ = Attachment Style Questionnaire; ANX = attachment anxiety; AVD = attachment avoidance; ESS = Experience of Shame Scale; CoSS = Compass of Shame Scale; AO = attack other; AS = attack self; WD = withdrawal; AV = avoidance; K10 = Kessler psychological distress scale. * *p* < 0.01. ** *p* < 0.001.

**Table 1 ijerph-20-03193-t001:** Means, standard deviations and Pearson’s correlations (*n* = 978).

Variables	Mean (SD)	1	2	3	4	5	6	7	8	9	10	11
1. Gender		-										
2. Age	32.17 (13.48)	−0.100 **	-									
3. Education		0.139 **	−0.198 **	-								
4. Job status		0.192 **	−0.165 **	0.033	-							
5. ASQ ANX	47.51 (12.18)	0.135 **	−0.197 **	0.056	0.068 **	-						
6. ASQ AVD	51.24 (10.74)	−0.058	−0.008	−0.054	0.004	0.368 **	-					
7. ESS	45.86 (14.02)	0.198 **	−0.302 **	0.119 **	0.065 **	0.636 **	0.288 **	-				
8. CoSS AO	11.47 (7.34)	0.133 **	−0.135 **	0.092 **	0.014	0.499 **	0.311 **	0.513 **	-			
9. CoSS AS	19.14 (10.23)	0.210 **	−0.248 **	0.130 **	0.104 **	0.595 **	0.238 **	0.658 **	0.530 **	-		
10. CoSS WD	15.45 (9.53)	0.232 **	−0.187 **	0.085 **	0.078 *	0.602 **	0.310 **	0.649 **	0.651 **	0.736 **	-	
11. CoSS AV	16.78 (7.16)	0.053	−0.201 **	0.034	0.050	0.273 **	0.281 **	0.351 **	0.509 **	0.413 **	0.436 **	-
12. K10	21.51 (7.43)	0.140 **	−0.159 **	0.065 *	0.073 *	0.555 **	0.304 **	0.641 **	0.404 **	0.545 **	0.529 **	0.217 **

Note. Gender was coded 0 = men, 1 = women. Education was coded 0 = high school degree, 1 = bachelors’ degree and above. Job status was coded 1 = employed, 0 = other. SD = standard deviation; ASQ = Attachment Style Questionnaire; ANX = attachment anxiety; AVD = attachment avoidance; ESS = Experience of Shame Scale; CoSS = Compass of Shame Scale; AO = attack other; AS = attack self; WD = withdrawal; AV = avoidance; K10 = Kessler psychological distress scale. * *p* < 0.01. ** *p* < 0.001.

**Table 2 ijerph-20-03193-t002:** Total, specific and serial indirect effects in the full model (*n* = 978).

Effects	β	SE	95% CI
Attachment anxiety			
Total indirect effect	0.336 **	0.025	[0.289, 0.379]
Specific indirect effects			
ASQ ANX→ESS→K10	0.226 **	0.023	[0.182, 0.263]
ASQ ANX→CoSS AO→K10	0.004	0.008	[−0.011, 0.019]
ASQ ANX→CoSS AS→K10	0.044 *	0.013	[0.020, 0.066]
ASQ ANX→CoSS WD→K10	0.020	0.012	[−0.004, 0.040]
ASQ ANX→CoSS AV→K10	−0.002	0.004	[−0.011, 0.003]
Serial indirect effects			
ASQ ANX→ESS→CoSS AO→K10	0.003	0.006	[−0.008, 0.013]
ASQ ANX→ESS→CoSS AS→K10	0.039 **	0.011	[0.019, 0.058]
ASQ ANX→ESS→CoSS WD→K10	0.017	0.011	[−0.003, 0.035]
ASQ ANX→ESS→CoSS AV→K10	−0.014 *	0.005	[−0.025, −0.007]
Attachment avoidance			
Total indirect effect	0.025	0.014	[−0.003, 0.053]
Specific indirect effects			
ASQ AVD→ESS→K10	0.030 *	0.011	[0.011, 0.052]
ASQ AVD→CoSS AO→K10	0.002	0.004	[−0.006, 0.011]
ASQ AVD→CoSS AS→K10	−0.001	0.004	[−0.009, 0.006]
ASQ AVD→CoSS WD→K10	0.005	0.004	[0.000, 0.015]
ASQ AVD→CoSS AV→K10	−0.017 *	0.006	[−0.031, −0.007]
Serial indirect effects			
ASQ AVD→ESS→CoSS AO→K10	0.000	0.001	[−0.001, 0.002]
ASQ AVD→ESS→CoSS AS→K10	0.005 *	0.002	[0.002, 0.011]
ASQ AVD→ESS→CoSS WD→K10	0.002	0.002	[0.000, 0.007]
ASQ AVD→ESS→CoSS AV→K10	−0.002 *	0.001	[−0.004, −0.001]

Note. *β* = standardized estimate; SE = standard error; CI = confidence intervals; ASQ = Attachment Style Questionnaire; ANX = attachment anxiety; AVD = attachment avoidance; ESS = Experience of Shame Scale; CoSS = Compass of Shame Scale; AO = attack other; AS = attack self; WD = withdrawal; AV = avoidance; K10 = Kessler psychological distress scale. * *p* < 0.01. ** *p* < 0.001.

## Data Availability

The data are available from the corresponding authors upon reasonable request.

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
