# Peer review of "A Serial Mediation Model of Insecure Attachment and Psychological Distress: The Role of Dispositional Shame and Shame-Coping Styles"

_ijerph, 2023, doi:10.3390/ijerph20043193_

Round 1
Reviewer 1 Report
The authors present a solid study that fills a significant gap in the literature. The manuscript is well-written. However, there are a few minor issues, which could easily be addressed. Most notably, I believe they have the correlations switch for two variables in Table 1.
The authors use “Withdrawn” rather than “withdrawal” occasionally.
Line 85: strongly related to psychopathy. First, I wouldn’t say strong (r = .27 and .37) with the two measures of psychopathy). Second, that comes from Reference 65, which isn’t listed here.
They might consider presenting the hypotheses as a path model.
The next two lines don’t really make sense:
“The scale dimensions were computed by summing up the number of items.”
“The scale was computed by summing up the number of items.”
“multivariate normality tests was used” – were used
Surprising correlations in Table 1 among CoSS scales and with ESS. AS/WD are usually highest among CoSS scales and with measures of shame, while AV is usually lowest among CoSS scales and with other measures of shame. These predictions are supported by the SEM in Fig 1. So, it looks like the rows for AV and WD could have been switched in Table 1. Alternatively, they could have computed the variables so that they added the items for AV but called it WD and vice versa.
“yielded to a good fit to the data” – delete “to”
“higher attachment avoidance was linked to higher avoidance shame-coping, and thus with higher psychological distress, with small effects” – I believe this sentence is incorrect. Since the path between AV and K10 is negative, the sentence should read “lower psychological distress”. Table 2 supports my interpretation as the indirect effect is listed as -.017
“Although previous studies evidenced that, out of the four shame-coping styles, attack self and withdrawal had the highest association with psychological problems [15, 16, 38], we found that only attack self and avoidance were significantly associated with psychological distress.” Two problems with this sentence. First, this isn’t true of the zero-order correlations in Table 1, especially if they switched the AV/WD rows. Second, this sentence may only be true of the model due to collinearity, e.g., AS/WD and AV/AO are fairly highly correlated.
“Another possible explanation is that avoidance helps maintain a positive self-image and experience less psychological distress.” First, awkward wording regarding less distress, maybe “helps people maintain … and to experience less…” Second, a third explanation is one they mentioned above – AV is the hardest style to measure via self-report because you are asking people to self-report their denial / emotional avoidance.
“Altogether, the incongruence of findings in relation to differing shame-coping styles not only validate shame as a multidimensional construct, but additionally proffers important questions on the different impact of shame-coping styles on mental health outcomes.” Incongruence here is ambiguous – with previous findings (don’t think this is what they mean). I think they mean differential pattern between the four styles and ESS / K10. Finally, “different impact” should be “differential impacts.”
Line 422 & 423: “shame-coping stile” - style
Line 431: “heighten distress” – heightened
Author Response
Reviewer 1
The authors present a solid study that fills a significant gap in the literature. The manuscript is well-written. However, there are a few minor issues, which could easily be addressed. Most notably, I believe they have the correlations switch for two variables in Table 1.
Response: We thank the Reviewer for his/her thoughtful comments, which helped us to strengthen the manuscript.
The authors use “Withdrawn” rather than “withdrawal” occasionally.
Response: We amended these typos.
Line 85: strongly related to psychopathy. First, I wouldn’t say strong (r = .27 and .37) with the two measures of psychopathy). Second, that comes from Reference 65, which isn’t listed here.
Response: We amended this sentence, stating that the correlation was moderate, and citing the paper from which this information came from. See line 111.
They might consider presenting the hypotheses as a path model.
Response: We thank the reviewer for this suggestion. We now added a figure of the hypothesized associations at the end of the introduction.
The next two lines don’t really make sense:
“The scale dimensions were computed by summing up the number of items.”
“The scale was computed by summing up the number of items.”
“multivariate normality tests was used” – were used
Response: We amended the sentences. See lines 205-207, 227-228, and 236-237.
Surprising correlations in Table 1 among CoSS scales and with ESS. AS/WD are usually highest among CoSS scales and with measures of shame, while AV is usually lowest among CoSS scales and with other measures of shame. These predictions are supported by the SEM in Fig 1. So, it looks like the rows for AV and WD could have been switched in Table 1. Alternatively, they could have computed the variables so that they added the items for AV but called it WD and vice versa.
Response: We truly thank the reviewer for spotting this. When preparing the Tables (including those in the supplementary materials) we listed the wrong name of most of the CoSS subscales. We now amended the Tables, and can confirm that CoSS AV has the lowest association with shame.
“yielded to a good fit to the data” – delete “to”
Response: we amended the sentence as suggested. See lines 305-306.
“higher attachment avoidance was linked to higher avoidance shame-coping, and thus with higher psychological distress, with small effects” – I believe this sentence is incorrect. Since the path between AV and K10 is negative, the sentence should read “lower psychological distress”. Table 2 supports my interpretation as the indirect effect is listed as -.017
Response: We thank the reviewer for this comment. We now amended the sentence. See line 323.
“Although previous studies evidenced that, out of the four shame-coping styles, attack self and withdrawal had the highest association with psychological problems [15, 16, 38], we found that only attack self and avoidance were significantly associated with psychological distress.” Two problems with this sentence. First, this isn’t true of the zero-order correlations in Table 1, especially if they switched the AV/WD rows. Second, this sentence may only be true of the model due to collinearity, e.g., AS/WD and AV/AO are fairly highly correlated.
Response: Thank you for this important consideration. We amended that sentence and limited our statement to the results of SEM, see lines 405-406. We have also added information on VIF and Tolerance values in the results. We found no evidence of multicollinearity between the predictors, which suggests that that individual pathways within our mediation model can be confidently interpreted. See lines 247-249 and 288-291.
“Another possible explanation is that avoidance helps maintain a positive self-image and experience less psychological distress.” First, awkward wording regarding less distress, maybe “helps people maintain … and to experience less…” Second, a third explanation is one they mentioned above – AV is the hardest style to measure via self-report because you are asking people to self-report their denial / emotional avoidance.
Response: We thank the reviewer for this insightful comment. We amended the sentence and also provided a third explanation to this finding, as suggested. See lines 417-420.
“Altogether, the incongruence of findings in relation to differing shame-coping styles not only validate shame as a multidimensional construct, but additionally proffers important questions on the different impact of shame-coping styles on mental health outcomes.” Incongruence here is ambiguous – with previous findings (don’t think this is what they mean). I think they mean differential pattern between the four styles and ESS / K10. Finally, “different impact” should be “differential impacts.”
Response: Thank you for this suggestion. We now revised this sentence, see lines 420-423.
Line 422 & 423: “shame-coping stile” – style
Response: We amended these typos. See lines 439 and 440.
Line 431: “heighten distress” – heightened
Response: We amended this typo, see lines 447-448.
Reviewer 2 Report
In this study, the authors attempted to study the "a serial mediation model of insecure attachment and psychological distress", where they thought that the dispositional shame and shame-coping styles play the mediating roles. This is an interesting study and the authors provided sound evidence to support their hypotheses. However, some problems should be resolved before publishing it.
(1) In the "Introduction", the authors introduced many things related to shame in paragraph 1 to paragraph 5, but there are less contents explain the relationships between the insecure attachment and psychological distress. Thus, I thought that the authors should firstly explain the relationships between the insecure attachment and psychological distress, then analyse whether dispositional shame and shame-coping styles can play the mediating roles.
(2) In addition, though the authors introduce lots of things related to shame, we did not know whether the dispositional shame and shame-coping styles can play the mediating roles due to the as above reasons.
Author Response
Reviewer 2
In this study, the authors attempted to study the "a serial mediation model of insecure attachment and psychological distress", where they thought that the dispositional shame and shame-coping styles play the mediating roles. This is an interesting study and the authors provided sound evidence to support their hypotheses. However, some problems should be resolved before publishing it.
(1) In the "Introduction", the authors introduced many things related to shame in paragraph 1 to paragraph 5, but there are less contents explain the relationships between the insecure attachment and psychological distress. Thus, I thought that the authors should firstly explain the relationships between the insecure attachment and psychological distress, then analyse whether dispositional shame and shame-coping styles can play the mediating roles.
Response: We thank the Reviewer for this comment. We now start the manuscript describing attachment theory, attachment insecurity and its association with psychological distress. We then make a small argument on the importance of studying potential mediators of these associations, to finally introduce shame and shame-coping styles. See lines 34-58.
(2) In addition, though the authors introduce lots of things related to shame, we did not know whether the dispositional shame and shame-coping styles can play the mediating roles due to the as above reasons.
Response: Thank you for this consideration. We have now clarified the potential mediating role of shame and shame-copying styles in the attachment insecurity-psychological distress relationship before introducing shame and shame-coping. See lines 55-58.

Reviewer 3 Report
This manuscript enlightens our understanding of shame in several ways, the most important of which is in connecting shame to problems of attachment. This link has been examined in some previous research but the present manuscript amplifies these connections, and offers a model for understanding how shame and attachment are related.
The study is rather simple in design, with no manipulations. Nevertheless, the authors are appropriately cautious in generalizing from their correlational/cross-sectional data, refraining from overgeneralizing. The mediation model presented in Figure 1 outlines a very plausible set of relationships among the variables. The next step would be to introduce either a longitudinal prospective design, or to consider manipulations of the variables at various points in the model in order to see the effect of increased/decreased values on the model.
I do not have any specific recommendations for this manuscript. In my opinion it is well-conceived and well-executed. The study is modest in design, and the authors are appropriately modest in describing its contribution. It is unusual that I do not see any changes that need to be made, but this manuscript seems acceptable basically as it is. It advances our understanding of shame and attachment. I recommend its publication.
Author Response
Reviewer 3
This manuscript enlightens our understanding of shame in several ways, the most important of which is in connecting shame to problems of attachment. This link has been examined in some previous research but the present manuscript amplifies these connections, and offers a model for understanding how shame and attachment are related.
The study is rather simple in design, with no manipulations. Nevertheless, the authors are appropriately cautious in generalizing from their correlational/cross-sectional data, refraining from overgeneralizing. The mediation model presented in Figure 1 outlines a very plausible set of relationships among the variables. The next step would be to introduce either a longitudinal prospective design, or to consider manipulations of the variables at various points in the model in order to see the effect of increased/decreased values on the model.
I do not have any specific recommendations for this manuscript. In my opinion it is well-conceived and well-executed. The study is modest in design, and the authors are appropriately modest in describing its contribution. It is unusual that I do not see any changes that need to be made, but this manuscript seems acceptable basically as it is. It advances our understanding of shame and attachment. I recommend its publication.
Response: We truly thank the Reviewer for his/her positive comments on our manuscript.
